# Exercise Therapy for Chronic Neck Pain: Tailoring Person-Centred Approaches within Contemporary Management

**DOI:** 10.3390/jcm12227108

**Published:** 2023-11-15

**Authors:** Rutger M. J. de Zoete

**Affiliations:** School of Allied Health Science and Practice, The University of Adelaide, Adelaide, SA 5005, Australia; rutger.dezoete@adelaide.edu.au

**Keywords:** neck pain, chronic pain, physical exercise, exercise therapy

## Abstract

Exercise therapy is considered the best evidence-based approach for managing chronic neck pain. However, the implementation of exercise therapy presents several challenges. Systematic reviews indicate that it has modest effectiveness, while clinical practice guidelines offer limited guidance on the exercise parameters required to optimise clinical outcomes. Moreover, recommendations often fail to differentiate between different types of neck pain. This article addresses the challenges associated with the prescription of exercise for chronic neck pain and provides recommendations for exercise therapy specific to chronic nociceptive, neuropathic, and nociplastic neck pain. The goal of this article is to facilitate the implementation of high-value evidence-based exercise therapy for these distinct types of chronic neck pain with the aim to improve its outcomes and to reduce the related individual and societal burdens.

## 1. Introduction

Neck pain is the fourth leading cause of years lived with disability globally [1]. Up to 70% of the global population will experience neck pain at least once in their lives [1,2] of which 50–85% is expected to become chronic within five years [1]. Chronic neck pain is responsible for a growing number of disability adjusted life years [3], contributing to a rapidly increasing trend in health care expenditures [4].

Neck pain is generally classified as either acute, which refers to pain that started less than 12 weeks ago, or chronic, which describes pain that has been present for 12 weeks or longer [5]. According to the International Association for the Study of Pain (IASP) classification of chronic pain phenotypes, chronic neck pain can be attributed to one of three mechanistic pain descriptors: nociceptive, neuropathic, or nociplastic [6]. Nociceptive pain arises from actual or potential tissue damage, neuropathic pain is the result of a lesion or disease of the nervous system, and nociplastic pain is caused by altered nociception without tissue damage [7].

Regardless of the type of chronic neck pain, current international clinical guidelines recommend conservative therapy as the primary treatment option, including physical exercise therapy; psychological treatment; and reassurance, advice, and education [8]. Of these, exercise therapy is commonly considered the mainstay treatment [9,10,11].

Despite its promise, however, the delivery of exercise therapy comes with its challenges. Several systematic reviews have concluded that the effectiveness of exercise therapy in improving pain intensity and pain-related disability is modest at best [12,13,14,15]. In addition to barriers such as patient compliance, expectations, preferences, and cost [16], clinicians experience challenges when prescribing exercise therapy for chronic neck pain. For example, whilst clinical guidelines recommend that exercise therapy be used as the primary treatment [9,10,11], the type, dosage, and intensity required to achieve optimal results is unclear. This leaves it up to the clinician prescribing the exercise therapy to use their own judgement, for example, having to choose out of numerous different types of physical exercise and having to determine patient-appropriate exercise parameters. This judgement will likely vary considerably between clinicians depending on their level of experience, expertise, and training [17]. Combined, the challenges associated with the prescription of exercise therapy for individuals experiencing chronic neck pain potentially limit the delivery of high-value care, leading to suboptimal clinical outcomes.

One way to facilitate decision making for exercise prescription is through tailoring the program to pain phenotypes. With reference to the IASP classifications for chronic pain, guidelines for the conservative management of neck pain do not differentiate between phenotypes, which means clinicians need to determine whether exercise therapy is appropriate for the patient’s pain presentation. In other words, how should the prescription of exercise differ between nociceptive, neuropathic, and nociplastic chronic neck pain presentations? 

This article aims to facilitate the clinical implementation of evidence-based exercise therapy to enhance the delivery of high-value care for chronic neck pain. By discussing and responding to the current challenges associated with exercise therapy, specific recommendations for its prescription are made for chronic nociceptive, neuropathic, and nociplastic neck pain.

## 2. Challenges in the Prescription of Exercise Therapy for Chronic Neck Pain

The prescription of exercise therapy for chronic neck pain comes with several challenges. Its complexities include the modest effectiveness of exercise therapy, the differential effects of various types of exercises, the varying responses of patients to exercise therapy, and the different types of chronic neck pain.

### 2.1. Modest Effectiveness of Exercise Therapy

Numerous systematic reviews and meta-analyses [12,13,14,15] have demonstrated that exercise therapy is modestly effective when compared to usual care or no treatment. The improvements in pain intensity and pain-related disability typically yield effect sizes ranging from 0.5 to 1.0. For example, the most recent Cochrane review on exercise for chronic mechanical neck pain reports a standardised mean difference of −0.71 for pain intensity when comparing strength-specific training with a control group [12]. This reduction in pain intensity equates to an approximate 10–15% improvement in self-reported pain intensity (1–1.5 points on a 0–10 numeric rating scale). A more recent meta-analysis [18] including different types of exercise therapy found larger effect sizes, for example, 1.32 for motor control training (2.6/10 improvement in pain intensity) and 1.21 for strengthening exercises (2.4/10 improvement). It is worthwhile to acknowledge that, although the overall effects reported in systematic reviews are modest, other approaches (e.g., nonsteroidal anti-inflammatory medication) yield similar efficacies [19]. Whilst modest effect sizes are indicative of possible clinical improvements, it must be noted that not all patients with chronic neck pain experience the same level of improvement. Also, there is substantial variability between trials investigating the effectiveness of exercise therapy for neck pain, which limits our ability to generalise their findings and pool the results. For example, an RCT investigating isometric strength training [20] reported an improvement of 40 points on a 0–100 VAS, whereas an RCT investigating a combination of endurance, strength, and range-of-motion exercises reported no treatment benefits [21]. The factors contributing to this variability include differences in the types of exercises assessed (isolated exercises or combinations), the inclusion of patients with acute or chronic pain, variations in intervention duration and intensity, and differences in the settings where the intervention was delivered.

### 2.2. Differential Effects of Different Types of Exercise Therapy

Traditional randomised controlled trials and meta-analyses often compare a single type of exercise to a control group. Alternatively, various types of exercises combined into one program are compared to a control group. While these studies generally indicate that exercise is modestly effective compared to no treatment or a control group, they provide little information about which type of exercise is most beneficial for improving patient outcomes. Clinical guidelines also provide only limited guidance on the recommended types of exercise therapy.

Some randomised controlled and clinical trials have directly compared the effectiveness of different types of exercise therapy [22,23,24,25,26,27]. However, the results of these comparisons have been mixed. For example, yoga and qigong were found to be more effective than neck-specific exercises for nonspecific neck pain, craniocervical flexion training was more effective than general strengthening training, and aerobic exercises were more effective than strengthening exercises for chronic whiplash-associated disorder (WAD). As such, evaluating the comparative effectiveness of different exercise types based on independent trials is challenging.

Recently, a network meta-analysis of exercise interventions for chronic nonspecific neck pain has enabled the comparison and ranking of different types of exercises [18]. While traumatic neck pain was excluded due to limited data, the study included 40 trials with over 3100 participants, investigating 11 types of exercise therapy for chronic nonspecific neck pain. The results indicated that motor control, mind–body exercises (e.g., yoga, Pilates, and tai chi), and strengthening exercises were the most effective in reducing pain intensity and pain-related disability.

Limited knowledge exists regarding the optimal prescription parameters for exercise therapy in the management of chronic neck pain. The establishment of a clear dose–response relationship, which defines the appropriate “dose” of exercises necessary to elicit the desired clinical response, remains elusive. Clinical guidelines offer limited guidance on the parameters to be considered when prescribing physical exercise therapy. Some research indicates that there may be minimal discernible differences in clinical outcomes when using different resistance training parameters [28]. Contradictory findings suggest that the dosage of exercise therapy is inversely correlated with neck pain intensity and pain-related disability, implying that a higher volume of exercise may yield better results [29]. It should be noted that these insights are based on two studies investigating neck, shoulder, and upper-extremity training, one small (*n* = 30) study with unknown exercise intensity [28] and a larger (*n =* 180) study with exercise intensity reported in metabolic equivalents (METs). This illustrates the current lack of comprehensive data, which hampers the ability to make informed clinical decisions regarding the parameters for exercise prescription, subsequently hindering its optimal integration into the treatment of chronic neck pain.

### 2.3. Differential Effects of Exercise Therapy across Patients with Chronic Neck Pain

One important observation is that exercise therapy is effective for some people, but it is not effective for others. It is possible that randomised trials may overlook these differential effects, as the improvements seen in responders are offset by nonresponders. For instance, if 50% of participants in a trial experience significant improvement (e.g., an effect size of 1.0) while the remaining 50% show no change (e.g., an effect size of 0), the average improvement across all participants appears modest, with an overall effect size of 0.5. For most reported randomised controlled trials, this hypothetical example likely holds true to some extent [30] due to factors such as mixed participant cohorts (acute, chronic, nonspecific, and WAD), the use of different types of physical exercises (e.g., strengthening, aerobic, and motor control), the setting (group versus individual sessions), and differences in adherence. 

Recent evidence supports the notion that exercise therapy may be effective for some patients but ineffective or even detrimental for others, with potential nonresponder rates of up to 50% in clinical trials [30]. Whilst numerous factors may contribute to this (e.g., the lack of adherence and variability in exercise programs), a proportion of people with neck pain may experience worsened pain following exercise therapy. Indeed, in a recent study [31] with high adherence to the exercise program, just over half of the participants (*n* = 13/24 (54%)) were found to experience clinically meaningful improvements (>20% improvement in pain-related disability). Whilst these findings are yet to be replicated, it is known that people with neck pain due to fibromyalgia are more likely to experience worsening of symptoms after exercise due to increased reactivity [32]. This not only affects the outcomes of randomised controlled trials but also emphasises the importance of identifying patients who may experience aggravated symptoms when prescribing exercise therapy. 

Single-case experimental design (SCED) studies have revealed the varied effects of exercise therapy among participants [24,33]. SCEDs, while typically being smaller studies that may not achieve the same level of statistical power and control as large-scale randomised trials, hold robust intrinsic value in investigating therapy effects in individual cases, offering insights and avenues for further exploration. Designed to assess the within- and between-participant responses, SCEDs have highlighted the diverse effects that exercise therapy can have on patients with chronic neck pain. The reasons why some individuals respond positively while others do not remain unclear; however, there may be different underlying mechanisms contributing to these differential responses. Central nervous system changes in patients with chronic neck pain could potentially explain these variations, as suggested by a recent study that found different brain changes in responders and nonresponders after exercise therapy [31]. This raises the questions of whether there are clinical subgroups among people with chronic neck pain and whether their response to exercise therapy is related to these subgroups.

### 2.4. Types of Chronic Neck Pain

Chronic neck pain typically stems from one of three main causes: (1) traumatic neck pain, such as WAD; (2) nonspecific neck pain (also referred to as idiopathic, insidious onset, or nontraumatic); or (3) neck pain resulting from specific pathologies like radiculopathy, spondylolysis, cancer, or osteoarthritis. These different types of neck pain exhibit significant differences in neurological mechanisms [34], physical and psychological symptoms [35], rehabilitation trajectories [36], and the impact of compensation claims [37].

Research often provides exercise therapy recommendations tailored to specific types of neck pain. For example, one study might investigate the effects of exercise for patients with WAD [38,39,40], while another study investigates the effects of exercise for patients with nonspecific neck pain [12,18,22,41]. Notably, the recommendations made for these different types of neck pain across studies do not appear to be all that different. This is further reflected by clinical practice guidelines that do not differentiate between the types of neck pain nor provide advice on tailoring exercise therapy prescriptions.

The comparison between traumatic and nontraumatic neck pain is commonly made, both in research and clinical practice. Indeed, stark differences in patient presentations have been observed. These differences encompass self-reported outcomes (such as pain-related disability and quality of life) [42,43], neck muscle function and head steadiness [42,44], pain sensitivity [42], brain structure (e.g., the degree of grey and white matter deficits) [34,45], changes in endogenous pain inhibition and conditioned pain modulation [34,43], and neuropsychological functioning [46]. It is commonly accepted that these factors should be considered during subjective and objective patient assessments and that they are likely to influence the management approach. 

In cases where chronic neck pain is accompanied by widespread hypersensitivity, the pain’s onset (e.g., traumatic versus non-traumatic) may be less significant when prescribing exercise therapy. Based on a patient’s presentation (assessed through a thorough subjective and objective assessment), a person-centred management plan can be tailored, for example, by placing emphasis on psychological interventions, education, or reassurance. Similarly, the patient presentation could be considered to inform the prescription of exercise therapy. Both traumatic and nontraumatic chronic neck pain may be the result of altered nociception, and exercise therapy should be tailored accordingly. Thus, instead of considering the pain’s onset, using the IASP classification for chronic pain (nociceptive, neuropathic, or nociplastic) may prove more helpful in informing exercise therapy prescription decisions.

### 2.5. Chronic Nociceptive Neck Pain

Chronic nociceptive neck pain arises from actual or potential tissue damage with nociceptive activation. Nociceptors are stimulated by thermal, mechanical, or chemical noxious stimuli, transmitting action potentials via Aδ and C primary sensory nerve fibres [47]. These fibres synapse in the dorsal horn of the spinal cord, where modulatory input from interneurons and descending pathways may inhibit or facilitate the transmission of the action potentials to higher cortical areas, resulting in the experience of pain [48]. An example of chronic nociceptive pain is osteoarthritis. Nociceptive pain is characterised by its local distribution (pain in the injured area, potentially with referred pain) and its provocation by movement or palpation.

### 2.6. Chronic Neuropathic Neck Pain

Chronic neuropathic neck pain is caused by a lesion or disease of neural tissue, with patients reporting pain in a neuroanatomically recognisable distribution, evidence of sensory dysfunction, and/or the diagnostic confirmation of a nervous system lesion [49]. Among various mechanisms driving neuropathic pain, two worth noting are the increased excitability of the dorsal root ganglion neurons due to changes in ion channels and neuroinflammation. Clinically, in addition to the neuroanatomically distributed pain pattern, neuropathic pain is recognisable by patients’ descriptions of electric-shock-like, shooting, or burning pain. Examples of chronic neuropathic neck pain include entrapment neuropathies (e.g., painful radiculopathy with radicular arm pain) and painful polyneuropathies (e.g., diabetic neuropathy and chemotherapy-induced neuropathy) [50,51].

### 2.7. Chronic Nociplastic Neck Pain

Chronic nociplastic neck pain arises from altered nociception without actual or threatened tissue damage. The IASP criteria for nociplastic pain include pain that is chronic (>3 months), regional in distribution, and not explained by nociceptive or neuropathic mechanisms; a history of hypersensitivity in the painful region; the presence of fatigue, sleep disturbance, cognitive problems, or increased sensitivity to stimuli (such as sound, light, or odors); and the elicitation of hypersensitivity in the painful region, manifested as allodynia or painful after-sensations following mechanical or thermal stimuli [7].

A key mechanism underlying nociplastic pain is believed to be pain sensitisation, which can occur centrally or peripherally. It is important to note that, while nociplastic pain and central sensitisation are often used interchangeably, they do not refer to the same constructs. Central sensitisation refers to the neurophysiological phenomenon of hyperexcitability in the central nociceptive pathways, resulting in increased sensitivity. Prolonged nociceptive input can lead to increased membrane excitability, facilitated synaptic function, and reduced inhibition, ultimately enhancing sensitivity to thermal, mechanical, or chemical noxious stimuli. Central sensitisation cannot be measured in humans; however, it is suggested that tests such as conditioned pain modulation, temporal summation, and quantitative sensory testing may allude to its presence [52]. The term nociplastic pain is a clinical descriptor used to identify patients whose symptoms align with the four criteria outlined in the IASP classification.

## 3. Prescribing Exercise Therapy for Different Types of Chronic Neck Pain

When using exercise therapy for the management of chronic neck pain, the program needs to be tailored to the individual. While considering the physiological responses to exercise, exercise recommendations can be provided for each pain phenotype (nociceptive, neuropathic, and nociplastic).

### 3.1. Physiological Effects of Exercise for Chronic Neck Pain

Depending on the type of exercise administered, a range of physiological effects can be elicited, both localised to the affected area and throughout the entire body. These effects might collectively contribute to pain relief. For example, exercise promotes improved blood flow, increasing the circulation of blood to the affected area. This enhanced blood flow delivers a higher concentration of oxygen and essential nutrients to the cervical tissues [53] while simultaneously aiding in the removal of waste products [54]. This effect is instrumental in reducing inflammation and promoting tissue healing [55], thereby addressing the underlying causes of pain. Another crucial aspect of exercise therapy is the strengthening of the cervical musculature. Exercise may enhance muscular strength, providing better support to the cervical structures and potentially contributing to diminishing pain [56]. Exercise also contributes to the production of synovial fluid, which has been shown to improve joint function in individuals with osteoarthritis [57], which may reduce joint pain and stiffness, ultimately enhancing overall mobility and potentially reducing discomfort.

Exercise also has the capacity to regulate neurotransmitter levels, including serotonin and norepinephrine, both of which play vital roles in pain perception. The balance achieved in neurotransmitter levels through exercise aids in modulating pain signals [58]. One well-recognised effect of exercise is the release of endorphins that bind to opioid receptors in the brain, leading to a reduction in pain [59]. In addition to these effects, exercise-induced endogenous pain inhibition is a noteworthy phenomenon. Exercise not only reduces the expression of serotonin transporters but also elevates serotonin levels in the brain, creating an environment conducive to pain relief [60]. Furthermore, exercise stimulates the release of endogenous opioids within the central inhibitory pathways, further enhancing the body’s natural pain-modulating mechanisms [60]. 

Although the physiological effects of exercise have been widely described in healthy populations, there are likely to be differences across people with chronic neck pain. In fact, evidence has shown that the differential effects of different types of exercise and the differences between patients may possibly be related to their differential physiological effects [31,38,61]. Some people with chronic neck pain may respond to exercise therapy with a decrease in pain sensitivity, known as exercise-induced hypoalgesia (EIH). However, others doing the same exercises may not experience the same decrease in pain sensitivity. Such findings indicate that exercise may not induce the same central endogenous pain modulation in all patients with chronic neck pain [39]. Studies investigating exercise-induced hypoalgesia have indeed shown that some people with chronic neck pain may have an impaired or absent hypoalgesic response, and in some cases, people even experience exercise-induced hyperalgesia (i.e., increased pain sensitivity after exercise), the opposite effect [62]. 

Exercise-induced effects are being increasingly studied in chronic neck pain. It is known that there are different neurobiological responses between different types of neck pain, for example, chronic WAD versus chronic nonspecific neck pain [34]. Additionally, different types of exercise have been shown to have different hypoalgesic effects [39]. Recently, a meta-analysis of all studies investigating the effects of exercise on EIH in chronic neck pain found that motor control exercises had the largest hypoalgesic effect [63]. Given the low-level intensity of motor control exercises, this likely contrasts the common beliefs around the required exercise intensity.

As pain sensitivity appears to change differently across patients with chronic neck pain, it seems that the mechanisms underlying such responses may differ accordingly. Because these sensitivity changes are typically related to the central nervous system, it is likely that central neurophysiological mechanisms play a role in these differential responses to physical exercise therapy [61]. Indeed, a recent study investigating brain changes after physical exercise therapy found differential structural brain changes between responders and nonresponders to the exercise program [31]. Furthermore, structural differences were already present at baseline, further supporting the notion of differential central neurophysiological mechanisms.

Regardless of the type of chronic neck pain, exercise therapy is likely to be utilised as a means to improve quality of life through achieving one of two goals or both goals: (1) pain relief and/or (2) improving physical function. Physical exercise therapy can be useful in achieving increased quality of life for all patients with chronic neck pain; however, the reasons and mechanisms for this are different for patients presenting with chronic neck pain due to nociceptive, neuropathic, or nociplastic reasons. Below is a description of exercise therapy approaches for the different phenotypes of chronic neck pain according to the IASP classification. The aim is not to provide an extensive program with detailed exercises but rather to provide considerations for tailoring exercise therapy within a person-centred approach. This allows exercises to be tailored to individual preferences. For the purpose of making systematic and individualised decisions to inform exercise prescription, the FITT-VP (frequency, intensity, time, type, volume, and progression) principals can be considered [64].

### 3.2. Exercise Therapy for Chronic Nociceptive Neck Pain

Treating chronic nociceptive neck pain focuses on reducing or eliminating the nociceptive driver of pain [65]. Exercise therapy is one of the possible approaches to achieve this, and when exercises are prescribed, this goal should be considered in terms of choosing appropriate exercise parameters [66]. The exercise parameters can start at endurance training, focussed on high-frequency, low-intensity, and neuromuscular control exercises. Once nociception has improved (reduced), exercise therapy can then focus on restoring full functionality of the affected tissues through the scaling up of the related exercise parameters (e.g., implementing resistance-based exercises and reducing repetitions).

In situations where the nociceptive driver cannot be completely eliminated, as seen in certain types of chronic nociceptive neck pain, like osteoarthritis, exercise therapy can still play a valuable role in improving physical function. While conservative treatment may not directly resolve the underlying joint pathology, exercise therapy can assist with optimising functional movements [57,67]. Commencing with exercises characterised by low-intensity, high-frequency routines, the approach gradually evolves towards more resistance-based regimens. This progressive approach helps to maintain joint flexibility and to alleviate stiffness, and it subsequently contributes to pain relief [68].

### 3.3. Exercise Therapy for Chronic Neuropathic Neck Pain

Exercise therapy typically plays a secondary role in the treatment of neuropathic pain, as it does not directly address the underlying neurological lesion causing the pain. However, research indicates that exercise can effectively manage the intensity of neuropathic pain symptoms, making it a valuable component of comprehensive treatment [69]. It is considered a complementary approach that works in conjunction with other treatments targeting the neurological lesion, such as medication and invasive procedures.

Unlike exercise therapy for nociceptive pain, the primary objective of exercise therapy for neuropathic pain is not to eliminate the biological driver of pain. Instead, its focus is to reduce hyperalgesia and allodynia [70], two prominent features of neuropathic pain. Physical exercise can contribute to alleviating these symptoms through various mechanisms, for example, normalisation of the inflammatory state, endogenous pain modulation, and/or normalisation of the microglial activity [70]. By incorporating physical exercise, individuals with neuropathic pain can experience a decrease in pain intensity, including reduced reactivity, minimised pain interference, and potentially fewer flare-ups [70]. In turn, this can lead to less pain-related disability and an improved quality of life.

When determining the parameters for the prescription of exercise for neuropathic pain, it is imperative to prioritise safety and comfort. Neuropathic pain can be exacerbated by high-intensity exercises. Therefore, it is generally advisable to start with low-to-moderate-intensity activities [69]. It is worth noting that higher volumes of physical activity have been associated with reduced neuropathic pain in patients undergoing chemotherapy [71] and those with type 2 diabetes [72]. Whilst this further emphasises the potential benefits of exercise in managing neuropathic pain, considerable thought needs to be given to the challenge of introducing increasing volumes of physical activity without exacerbating symptoms. For instance, this could be achieved by engaging in exercises with low intensity, focusing on maximising the volume of the training program. These activities are less likely to trigger or worsen neuropathic symptoms [73].

### 3.4. Exercise Therapy for Chronic Nociplastic Neck Pain

Different from exercise therapy for nociceptive pain, when dealing with chronic nociplastic neck pain, exercise therapy is not used to remove the nociceptive driver. Rather, the focus of exercise therapy for nociplastic pain, which is often presented in conjunction with central sensitisation, is to reduce widespread hypersensitivity, hyperalgesia, and/or allodynia. Here, exercise therapy for nociplastic pain and central sensitisation can be used to improve the condition of the neurophysiological system. Central sensitisation is defined as the increased sensitivity of the nociceptive neurons of the central nervous system [74], and, as such, exercise therapy can be used as a means to facilitate a desensitisation process. In this process, gradual exposure to stimuli of increasing strength (in this case, exercise) can be used to decrease sensitivity [75]. 

There are currently no guidelines available for exercise therapy parameters that achieve desensitisation. However, the type of exercise may be tailored based on the patient’s and clinician’s preferences, considering the focus is to expand the volume of physical activity, which will lead to improved functional activity and engagement. Further recommendations can be taken from available research and clinical responses. According to investigations into the effects of exercise in patients with chronic fibromyalgia, it has been suggested that the best intensity for exercise therapy is the intensity preferred by the patient [76]. This work has shown that exercise therapy at the preferred intensity (around 45% of the maximum heart rate (MHR)) is more effective than exercise therapy at a higher intensity (>60% of MHR). 

When exercise parameters for chronic nociplastic neck pain are selected, low-intensity exercises could be an appropriate starting point. Opposed to high-intensity training, this feasible entry level will allow engagement with the training without exacerbating symptoms. Maximising the volume of the training program whilst keeping the intensity low is desirable. An important note here is to remember that the focus is not to improve the physical capacity but rather to desensitise and reduce pain sensitivity. Gradually increasing exercise intensity through graded exercise allows for sensitivity build-up. Starting at a low exercise intensity has the added benefit of reassuring patients about their ability to perform the exercises, thereby reducing fear avoidance behaviour.

If a patient exhibits high reactivity, characterised by a low pain threshold and increased pain hypersensitivity, hyperalgesia, and/or allodynia, it is advisable to initiate exercise therapy at a very low intensity. Examples of suitable approaches include range-of-motion exercises, motor control training, or aquatic therapy in a pool. If even these low-intensity exercises are too demanding for the patient (causing a flare-up), the clinician can consider alternative strategies to achieve some level of desensitisation, such as reassurance, advice, and education, prior to starting exercise therapy. This approach minimises the risk of triggering a flare-up in symptoms.

## 4. Exercise Therapy within Contemporary Pain Management

In contemporary pain management approaches, exercise therapy plays a crucial role. However, it should rarely be delivered in isolation, as patient presentations are often complex and multifactorial, requiring a holistic approach consistent with the biopsychosocial model. In these cases, a multimodal treatment approach is necessary to address the patient’s needs effectively.

### 4.1. Complex Presentations and Mixed Neck Pain Types

Theoretically, discriminating between pain types is useful for personalised exercise therapy prescription; however, clinically, patients rarely present with a single driver for pain that we can identify and fully isolate. Rather, pain presentations are often based on mixed pain types. One pain type might be dominant, but there may still be multiple mechanisms at play. For example, in a case of osteoarthritis of the cervical spine, chronic neck pain may be of nociceptive and nociplastic nature concurrently. Unfortunately, as a clinician it is often hard to assess the respective contributions to the pain experience. Therefore, phenotyping is more complex than simply assigning pain to one of three pain types, especially when providing convincing evidence of one pain type is difficult, for example, with nociplastic pain. Therefore, it is likely more effective to identify the “possible” and “probable” drivers for pain.

### 4.2. Exercise Therapy in the Biopsychosocial Context

It is impossible to state that exercise therapy is merely a physical treatment and that we should assess its value only by its physiological effects. Depending on its setting, delivery, and mode, exercise therapy addresses various components related to chronic neck pain. In addition to its physical effects, exercise therapy may have a positive impact on cognitive factors. For example, through exercise therapy and reflection on their responses (hypoalgesia versus hyperalgesia), a patient’s understanding of pain and movement is likely to improve. It is well established that psychological comorbidities, including depression, anxiety, and stress, may benefit from engagement in physical exercises [77]. Exercise therapy may also have positive effects on social aspects, as exercise therapy is often delivered in group sessions, for example, yoga, Pilates, and tai chi. These low-intensity mind–body types of exercise, often delivered as exercise classes, are amongst the most effective for improving pain and disability for patients with chronic neck pain [18]. Specifically, group composition, including factors such as the group size and group member characteristics, has been confirmed to have an impact on individual treatment outcomes [78]. Lastly, exercise therapy is likely to exert positive psychological effects through its central mechanisms, as most people feel happier and more positive after physical exercise [79].

Besides its broad positive effects when applied in isolation, exercise therapy is rarely utilised as a unimodal approach. Prescribing exercise therapy provides the opportunity to address other components of chronic neck pain, and it allows a clinician to deploy approaches such as cognitive behavioural therapy [80] and pain neuroscience education [81].

### 4.3. Different Responses to Exercise Therapy

Exercise therapy may not immediately improve someone’s pain condition, especially in cases of chronic neck pain, but it can have immediate short-lasting hypoalgesic effects. For some patients, exercise may not vastly improve the condition at all; however, in such cases, exercise can be used to prevent symptoms from getting worse and can mitigate deconditioning. At the same time, pain exacerbation is highly common after a single bout of exercise [60], and it is crucial not to contribute to catastrophising beliefs when short-term exercise-induced hyperalgesic responses occur, as they are common with exercise therapy [82,83]. Patients should be informed that it often takes time to see improvements. Commitment and adherence are crucial during this period and are likely to be challenged by factors such as exercise preference and expectations. 

Whilst some persistence is required to ensure that there truly is a nonresponse or worsening and not just initial short-lasting hyperalgesic effects, it is important to identify those patients who do not improve or might get worse through exercise therapy. As described under exercise therapy for chronic nociplastic neck pain, it is essential to recognise that the state of hypersensitivity in some patients may be so severe that exercise therapy could initially exacerbate their condition. In such cases, it is crucial to consider a holistic approach to pain management. This may involve a multidisciplinary strategy, including the biopsychosocial management of symptoms, to support patients and to enable them to engage optimally with other therapeutic approaches. Additionally, in cases of severe pain, other interventions may be necessary, including medication, anaesthesia, and other nonexercise-based physiotherapeutic interventions. It is common practice to begin with drug-free pain treatment methods before considering other options. Here, exercise therapy is still an important treatment to come back to considering its positive effects on physical capacity and quality of life. However, if exercise is not helpful or if it makes the condition worse, alternative treatment options could be explored.

### 4.4. The Future of Exercise Therapy for Chronic Neck Pain

Ultimately, the goal is to prescribe exercise therapy to those patients for whom it will be effective. At the same time, this means trialling other types of treatment for those patients for whom exercise therapy is not effective. Emerging evidence is becoming available suggesting methods to provide a meaningful prediction of the therapeutic effectiveness in chronic pain populations [31,84]. Improvements in prediction capabilities will enhance targeted and personalised pain medicine and will likely inform who would benefit most from exercise therapy for chronic neck pain.

## 5. Conclusions

The prescription of exercise therapy for chronic neck pain presents a complex challenge for clinicians. Numerous factors must be considered, and both clinicians and patients may have concerns about exacerbating pain. However, by gaining a deep understanding of the underlying mechanisms of exercise therapy and recognising the diverse goals for different neck pain phenotypes, clinicians may be able to enhance the effectiveness of exercise therapy in managing chronic neck pain. By tailoring exercise programs to individual needs, addressing barriers, and fostering a collaborative approach, we can optimise the delivery of exercise therapy and improve the outcomes for patients with chronic neck pain. Continued research and advancements in personalised pain medicine hold the potential to further refine the selection of appropriate candidates for exercise therapy, leading to better outcomes and enhanced quality of life for individuals living with chronic neck pain.

## Data Availability

Not applicable.

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
