# Peer review of "Exercise Therapy for Chronic Neck Pain: Tailoring Person-Centred Approaches within Contemporary Management"

_jcm, 2023, doi:10.3390/jcm12227108_

Round 1

Reviewer 1 Report

Comments and Suggestions for Authors

Exercise Therapy for Chronic Neck Pain: Tailoring Person-Centred Approaches within Contemporary Management

This is an extensive and well-written overview of the subject. It is important to consider in terms of the rehabilitation of various neck pain patients and that different therapy approaches is needed. Some clarifications and suggestions for improvement of the article are presented below.

Up to 70% of the population will experience neck pain at least once in their lives [1, 2], of which 50-85% is expected to become chronic within five years [1].

This means that prevalence of chronic neck pain would be 35 % i.e. more than every third person would have chronic neck pain? F.ex. in Finland the prevalence is 4 % in men and 6% in women (Aromaa A, Koskinen S, ed. Health and functional capacity in Finland. Baseline results of the Health 2000 health examination survey. Publications of the National Public Health Institute B3/2002. Helsinki 2002). Postal surveys are known to cause overestimates of prevalence and most studies are postal surveys. The frequency range chronic pain should be reported.

According to the International Association for the Study of Pain (IASP) classification of chronic pain phenotypes, chronic neck pain can be identified as being a result of nociceptive, neuropathic, or nociplastic mechanisms [6].

This classification misses psychosomatic pain, which should be included.

This reduction in pain intensity equates to an approximate 10-15% improvement in self-reported 77 pain intensity (1-1.5 points on a 0-10 numeric rating scale). Whilst this is not a very large improvement in itself…

The reader understands this to mean that the effectiveness of exercise therapy is weak. I would give more perspective to this as NSAID have been shown also to give 1 points pain reduction on a 0-10 numeric rating scale and opiates are not better, but give more side effects, which have to be considered with all medication. Moreover, one should not rely only on meta-analysis, which but all exercise interventions together without considering huge difference between them. Categorization of exercise type is often based on articles, which have often poorly written and thus not all exercise parameters have been properly considered (exercise type, load, intensity, frequency, length of training time). Thus I suggest that in addition to average improvement is to show how big improvement has been achieved in proper neck strength training studies.

Some research indicates that there may be minimal discernible differences in clinical outcomes when using different resistance training parameters [25]. Contradictory findings suggest that the dosage of exercise therapy is inversely correlated with neck pain intensity and pain-related disability, implying that a higher volume of exercise may yield better results [26].

Instead of directly comparing these studies as equal one should explain possible difference between “constradictory” findings. [25] n =32 (i.e a pilot study), training period 2 mo for upper arms (no neck exercise and the intensity was not told) [26] n =180, specific training 12 mo for neck muscles and exercise load individually measured and adapted. Basic exercise physiology does not recommend strength training twice a day like has been performed in study by Saeterbakken et al.  It may have negative impact to the results. Moreover, basic exercise physiology does recommend specific training i.e. training of neck muscles to improve neck function and not upper extremities – compare upper arm training for knee pain. There are several studies with more appropriate neck exercise methods: Ylinen et al. (2003), Chiu et al. (2005), Jull et al. (2007), Andersen et al. (2008 ja 2011), Aslan et al. (2011), Rendant et al. (2011), Bronfort et al. (2012), Borisuit et al. (2013), Zebis et al. (2014) Park et al. (2015) etc.

This raises the question whether there are clinical subgroups among people with chronic neck pain and whether their response to exercise therapy is related to these subgroups.

I agree about this opinion, but the majority of non-responders is due to low compliance, which is important to mention. Fibromyalgia is an important group that should be highlighted (5% of women and 1 % of men), because all have neck pain. Although exercise therapy has shown to be effective also for those, people tend to get more easily painful reaction about exercising although it varies periodically, which has to be taken in account.

Based on a patient’s presentation assessed through a thorough subjective and objective assessment, a person-centered management can be tailored, for example by placing emphasis on psychological interventions, education, or reassurance. Similarly, the patient presentation could be considered to inform the prescription of exercise therapy.

This sub-chapter is under whiplash, which may lead stigmatizing this patient group. However, most patients after whiplash recover without problems. I would rather recommend moving it under own chapter as people with all types of chronic neck pain may need psychological intervention. Moreover, psychosomatic neck pain may not benefit from exercise therapy at all and is not only waste of resources, but it may delay the introduction of the correct forms of therapy.

 Examples of chronic nociceptive pain include osteoarthritis, tendinosis, and bursitis.

Tendinosis is not a diagnosis in neck area and even tendinitis is very rare.  Bursitis is not a diagnosis in neck area. Whiplash injury may cause degenerative instability and pain. Joint stiffening due to inflammatory reaction in result of high energy trauma may be cause of neck symptoms.  

Exercise also contributes to improved joint function: through consistent movement during exercise, the production of synovial fluid is stimulated [53].

Indeed movement is essential for joints to function. However, there is no evidence that only moving the joint would improve chronic neck pain and mobility.  The reference is about the synovium theory of knee osteoarthritis and not about neck. On the contrary randomized studies have shown that low intensity exercises, as well as nonspecific training is ineffective (Viljanen et al 2003).  In contrast specific isometric strength exercises have shown to improve neck mobility significantly more than stretching alone or low intensity neck muscle training (Ylinen et al. 2003). Joints are not moved in isometric exercises, but deep neck muscles cause dynamic cyclic stretching on joint capsules as attachments of deep muscle cover about 25 % of the surface of joint capsules The resistance exercise with the load of 80% of maximum has shown to result improved ROM, and decrease pain and disability and increased pressure pain threshold (Ylinen et al. 2005). Decreased ROM is often an important factor causing pain, which is often omitted in studies as it has not been even measured.

Recent evidence supports the notion that exercise therapy may be effective for some patients but ineffective or even detrimental for others.

I do not agree with opinion that exercises studies show poor results, because only half of the patients get better, because other half get worse. There is no evidence about such phenomenon. Referred Karlsson et al. clearly state that poor results were due to lack of adherence and not due exercising. If patients get worse due to exercise therapy, it means side effect or complication, which should have been reported and I do not suspect that it had not been done with purpose.  The other reason to poor results is simply that the study was not at all proper resistance training study; the results show that there was no strength increase compared to the control group. This is common for many studies published, because exercise physiology has not been properly taught to healthcare professionals. Moreover, if exercise method improves half of patients and makes other half worse, it should be abandoned, if responders cannot be selected beforehand. This chapter needs to be rephrased or deleted.

Single-case experimental design (SCED) studies have revealed varying effects of exercise therapy among participants.

Case study is the lowest category of research, as it does not show evidence of therapy effect, as results may be due to many other factors. However, they may rise an idea that the therapy form could be worth studying. Please, rephrase the sentence.  Many exercise studies have too small number of subjects to make proper statistics and to control confounding factors, which mean that they are not much better than case studies.

As described under exercise therapy for chronic nociplastic neck pain, potentially the state of hypersensitivity is too severe, and other approaches need to be considered to lessen the reactivity. A multidisciplinary approach, including biopsychosocial management of symptoms, is important so that patients can engage optimally with other approaches.

Any type of pain may be so severe that exercise therapy initially aggravates it. Biopsychosocial management is important to support patient, but treatment of pain may also needed (medication, anesthesia, different pain treatments performed by physiotherapist) is important to mention. Commonly it is recommended to use drug free pain treatment methods first.

Reviewer 2 Report

Comments and Suggestions for Authors Dear authors: First of all, I would like to congratulate you on your research work. The topic of the manuscript is interesting, especially because, as you point out, neck pain in systematic reviews does not get evidence for therapeutic exercise, but in clinical recommendations it does, so I consider it interesting and I find it a challenge for authors and reviewers. I would like to suggest some areas of improvement for your article:   - You should use abbreviations for the term chronic neck pain (CNP) to make the manuscript easier to read.   - In my opinion, you omitted the pain types from line 155-189.   - The manuscript only has the introduction and conclusion sections and in my opinion the methods, results and discussion sections should be added.   - In addition, you comment that your article is a review, so you should add basic aspects of review articles.   In conclusion, the subject matter of the article is interesting, but in my opinion you should stick to the sections established by the publisher for publication and to the journal's regulations.   Regards.

Round 2

Reviewer 2 Report

Comments and Suggestions for Authors

Dear authors:

Thank you for your quick and organised response. After your explanation all the suggestions made are understood, I would just add abbreviations to the term chronic neck pain (CNP) please.

Thank you, best regards